# Hypoxia-enhanced Blood-Brain Barrier Chip recapitulates human barrier function and shuttling of drugs and antibodies

Tae-Eun Park [1,6,9], Nur Mustafaoglu[1,9], Anna Herland[1,7,8], Ryan Hasselkus [1], Robert Mannix[1,5], Edward A. FitzGerald [1], Rachelle Prantil-Baun[1], Alexander Watters[1], Olivier Henry[1], Maximilian Benz [1], Henry Sanchez[1], Heather J. McCrea[2], Liliana Christova Goumnerova [2], Hannah W. Song[3], Sean P. Palecek[3], Eric Shusta[3] & Donald E. Ingber [1,4,5]

The high selectivity of the human blood-brain barrier (BBB) restricts delivery of many pharmaceuticals and therapeutic antibodies to the central nervous system. Here, we describe an in vitro microfluidic organ-on-a-chip BBB model lined by induced pluripotent stem cell-derived human brain microvascular endothelium interfaced with primary human brain astrocytes and pericytes that recapitulates the high level of barrier function of the in vivo human BBB for at least one week in culture. The endothelium expresses high levels of tight junction proteins and functional efflux pumps, and it displays selective transcytosis of peptides and antibodies previously observed in vivo. Increased barrier functionality was accomplished using a developmentally-inspired induction protocol that includes a period of differentiation under hypoxic conditions. This enhanced BBB Chip may therefore represent a new in vitro tool for development and validation of delivery systems that transport drugs and therapeutic antibodies across the human BBB.

[1] Wyss Institute for Biologically Inspired Engineering at Harvard University, Boston, MA 02115, USA. [2] Department of Neurosurgery, Boston Children's Hospital and Harvard Medical School, Boston, MA 02115, USA. [3] Department of Chemical and Biological Engineering, University of Wisconsin-Madison, Madison, WI 53706, USA. [4] Harvard John A. Paulson School of Engineering and Applied Sciences, Harvard University, Cambridge, MA 02138, USA. [5] Vascular Biology Program and Department of Surgery, Boston Children's Hospital and Harvard Medical School, Boston, MA 02115, USA. [6] Present address: Ulsan National Institute of Science and Technology (UNIST), UNIST-gil 50, Ulsan 44919, Republic of Korea. [7] Present address: Division of Micro and Nanosystems, KTH Royal Institute of Technology, Stockholm, Sweden. [8] Present address: Swedish Medical Nanoscience Center, Department of Neuroscience, Karolinska Institute, Stockholm, Sweden. [9] These authors contributed equally: Tae-Eun Park, Nur Mustafaoglu. Correspondence and requests for materials should be addressed to D.E.I. (email: don.ingber@wyss.harvard.edu)

The human blood-brain barrier (BBB) is a unique and selective physiological barrier that controls transport between the blood and the central nervous system (CNS) to maintain homeostasis for optimal brain function. The BBB is composed of brain microvascular endothelial cells (BMVECs) that line the capillaries as well as surrounding extracellular matrix (ECM), pericytes, and astrocytes, which create a microenvironment that is crucial to BBB function[1]. The brain microvascular endothelium differs from that found in peripheral capillaries based on its complex tight junctions, which restrict paracellular transit and instead, require that transcytosis be used to transport molecules from the blood through the endothelium and into the CNS[2]. BMVECs also express multiple broad-spectrum efflux pumps on their luminal surface that inhibit uptake of lipophilic molecules, including many drugs, into the brain[3,4]. The astrocytes and pericytes provide signals that are required for differentiation of the BMVECs[5,6], and all three cell types are needed to maintain BBB integrity in vivo as well as in vitro[7,8]. The BBB is also of major clinical relevance because dysfunction of the BBB is observed in many neurological diseases, and the efficacy of drugs designed to treat neurological disorders is often limited by their inability to cross the BBB[9]. Unfortunately, neither animal models of the BBB nor in vitro cultures of primary or immortalized human BMVECs alone effectively mimic the barrier and transporter functions of the BBB observed in humans[10–12]. Thus, there is a great need for a human BBB model that could be used to develop new and more effective CNS-targeting therapeutics and delivery technologies as well as advance fundamental and translational research[7,8].

Development of human induced pluripotent stem (iPS) cell technology has enabled differentiation of brain-like microvascular endothelial cells (iPS-BMVECs) that exhibit many properties of the human BBB, including well-organized tight junctions, expression of nutrient transporters and polarized efflux transporter activity[13,14]. The trans-endothelial electrical resistance (TEER) values exhibited by the permeability barrier generated by these human iPS-BMVECs reach physiological levels ($\sim$3000–5000 $\Omega\cdot cm^2$) within 24–48 h when cultured in Transwell inserts or within a microfluidic organ-on-a-chip (Organ Chip) device[15–17], a level that is more than an order of magnitude higher than TEER values previously reported in other in vitro human BBB models[6,15]. However, the usefulness of these iPS-BMVEC models for studies on targeted delivery to the CNS is limited because they can only maintain these high TEER levels for $\sim$2 days, and the expression of efflux pumps in these iPS-BMVECs does not fully mimic those of human brain endothelium in vivo[18]. Here, we describe the development of an enhanced human BBB model created with microfluidic Organ Chip culture technology[19] that contains human iPS-BMVECs interfaced with primary human pericytes and astrocytes, and that uses a developmentally inspired differentiation protocol[20–22]. The resulting human BBB Chip exhibits physiologically relevant levels of human BBB function for at least 1 week in vitro, including low barrier permeability and expression of multiple efflux pumps and transporter functions that are required for analysis of drug and therapeutic antibody transport.

## Results

**Developmentally inspired differentiation of brain endothelium.** Given that the BBB first forms in the developing brain in a relatively oxygen-poor environment (1–8%) before establishment of the circulatory system[23], and oxygen availability has been shown to play a vital role in endothelial differentiation from a variety of stem cell sources[20,21], we hypothesized that culturing iPS cells under similar hypoxic conditions could generate more

highly differentiated BMVECs and potentially stabilize their phenotype. The published differentiation induction protocol for creating human BMVECs involves culture of iPS cells on Matrigel for 3 days in mTeSR1 medium followed by 6 days in DMEM-F12 medium, and then 3 days in endothelial cell medium containing retinoic acid under normoxic (20% $O_2$) conditions[13]. Thus, to explore this possibility, we modified this method for inducing differentiation of BMVECs from human iPS cells by shifting the cultures to hypoxic conditions (5% $O_2$) for the last 9 days of the induction protocol (Supplementary Figure 1a). Importantly, we found that exposure to hypoxia during this differentiation protocol produced significant (2–6-fold) increases in the mRNA levels for the endothelial cell–cell adhesion molecules, VE-cadherin (vascular endothelial cadherin) and PECAM-1 (platelet endothelial cell adhesion molecule, also known as CD31), as well as the influx transporter GLUT-1 (BBB-specific glucose transporter), efflux transporter P-gp (permeability glycoprotein), and VEGF-A (angiogenic vascular endothelial growth factor-A), relative to control iPS-BMVECs induced under normoxic conditions (Supplementary Figure 1b and Supplementary Table 1). The increased expression of these endothelial molecules that play an essential role in CNS neovascularization[24,25] confirmed the successful transition of the iPS cells into a human BMVEC phenotype.

Canonical Wnt/β-catenin signaling is essential for development of brain microvessels, and Wnt ligands Wnt7a and Wnt7b have been implicated in BBB development in vivo[16,26]. We similarly found that Wnt7a mRNA levels increased by over 25-fold with the hypoxic induction protocol compared with normoxic conditions, while Wnt7b levels remained unchanged (Supplementary Figure 1c and Supplementary Table 2). In addition, past in vitro and in vivo studies showed that Wnt/β-catenin signaling pathways interplay with hypoxia-induced factor 1α (HIF1α) signaling[27,28]. Indeed, ELISA analysis revealed that HIF1α protein levels were upregulated by exposure to hypoxia during the differentiation protocol; however, they decreased when the differentiation was completed and the cells shifted to normoxic conditions after seeding on-chip (Supplementary Figure 1d). Interestingly, we found that we could produce similar results by exposing the iPS cells to 100 μM cobalt chloride ($CoCl_2$) for the same 9-day period while under normoxic conditions (Supplementary Figure 1d); $CoCl_2$ has been previously shown to chemically mimic the effects of hypoxia by stabilizing HIF1α[29]. This finding may obviate the need for specialized gases and culture chambers for induction of improved BMVEC differentiation in future studies.

**Reconstitution of a human BBB on-chip under hypoxia.** To explore whether these hypoxia-induced iPS-BMVECs could be used to build a human BBB Chip with enhanced functionality, we used soft lithography to create optically clear, poly(dimethylsiloxane) (PDMS), 2-channel, microfluidic devices containing an upper CNS microchannel separated from a parallel vascular microchannel by a porous (2 μm diameter), polyethylene terephthalate (PET) membrane coated on both sides with an ECM composed of collagen type IV and fibronectin (Fig. 1a). A mixture of primary human brain astrocytes and pericytes (seeding ratio of 7:3) were then cultured in the astrocyte medium on the PET membrane in the upper channel under static conditions for 1 h. Human iPS-BMVECs cultured for 8 days under hypoxic (5% $O_2$) conditions on Matrigel were removed using Accutase and plated on the bottom surface of the ECM-coated porous membrane in the lower channel of the microfluidic chip in endothelial medium with addition of retinoic acid (RA) using similar culture conditions and flow rate. Chips were flipped and incubated for 5 h to induce adhesion of iPS-BMVECs to the membrane. The BBB

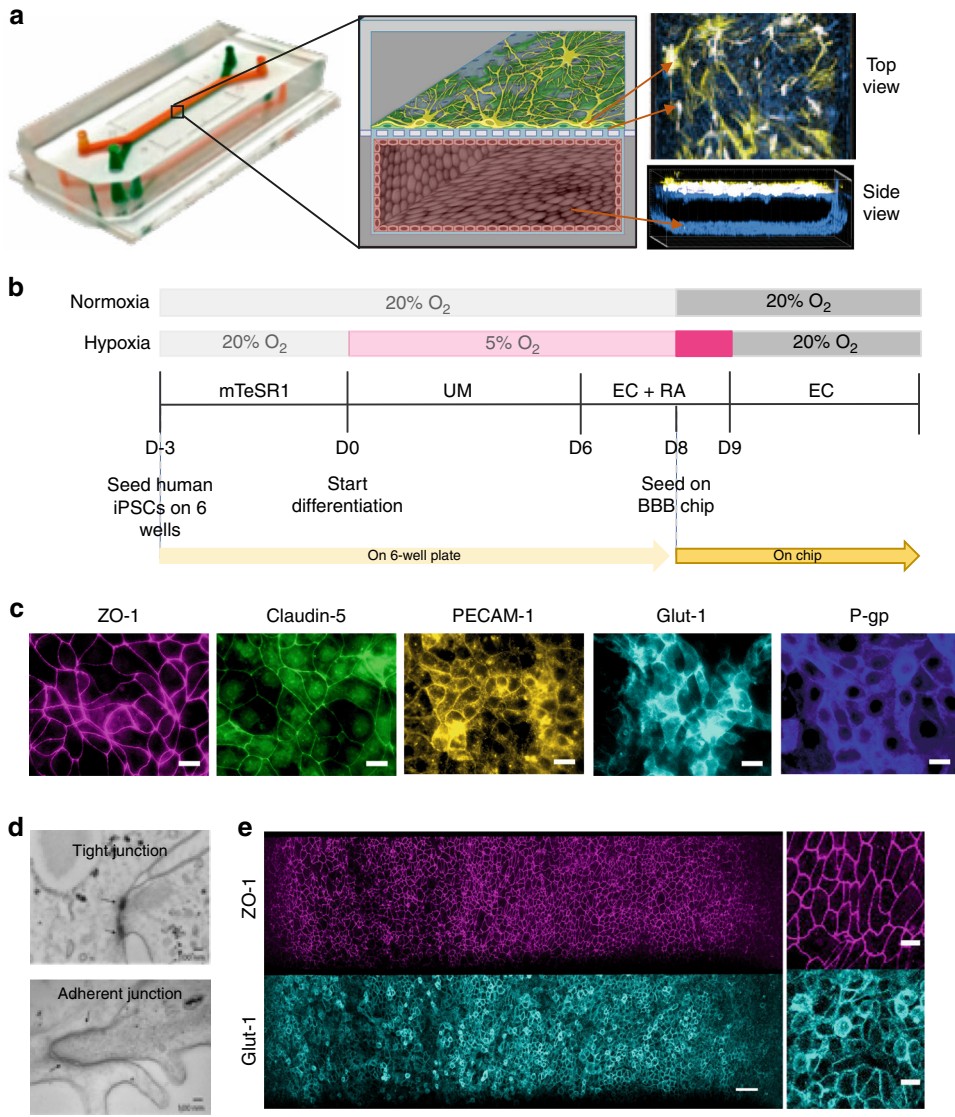

**Fig. 1** Reconstitution of the human BBB in an Organ Chip microfluidic device. **a** Photograph (left), schematic illustration (center), and immunofluorescence micrographs (right) of a 2-channel microfluidic Organ Chip with iPS-BMVECs cultured on all surfaces of the basal vascular channel, and primary human brain astrocytes and pericytes on the upper surface of the central horizontal membrane in the apical parenchymal channel. At the top right, z-stack images of the pericytes (yellow, F-actin staining) and astrocytes (white, GFAP staining) in the top channel of the BBB Chip are reconstituted and shown from above; a side view of similar stacked images for the lower vascular channel containing BMVECs (blue, ZO-1 staining) is shown at the bottom right. A video of overlay z-stack reconstituting this 3D BBB can be seen in Supplementary Movie 1. **b** Timeline for the in vitro differentiation of the human iPS-BMVECs, and seeding in the BBB Chips. **c** Immunofluorescence micrographs of the human brain endothelium cultured on-chip for 3 days labeled with ZO-1, Claudin-5, PECAM-1, GLUT-1, and P-glycoprotein (bar, 20 μm). **d** Electron micrographs of the human brain microvascular endothelium after 3 days in the BBB Chip, highlighting the presence of well-formed tight junctions (top, arrow) and adherens junctions (bottom, arrow) (bar, 100 nm). **e** Low (left) and high (right) magnification immunofluorescence micrographic views of the human brain microvascular endothelium cultured on-chip for 3 days viewed from above demonstrating high levels of expression of ZO-1 and GLUT-1 across the entire endothelial cell monolayer (bar, 50 μm)

Chip was cultured for one additional day under hypoxia to promote endothelial monolayer formation and to acclimate the cells to the device before being shifted to normoxic conditions for the remainder of the experiment and before initiating continuous medium flow ($100 \, \mu L \, h^{-1}$) (Fig. 1b). We also controlled flow through the vascular channel to maintain physiological levels of fluid shear stress ($6 \, dyne \, cm^{-2}$ at $100 \, \mu L \, h^{-1}$) and a blood-like viscosity (3–4 cP; modified by adding 3.5% dextran to the medium).

Confocal immunofluorescence microscopic analysis after 3 days of microfluidic culture revealed that these conditions resulted in formation of an iPS-BMVEC monolayer that covered all four walls of the lower channel, creating a hollow vascular lumen, in addition to being interfaced directly across the porous ECM-coated membrane with primary human pericytes and astrocytes in the CNS channel above (Fig. 1a, Supplementary Movie 1). Higher magnification views revealed that the astrocytes extended processes through the 2 μm pores of the PET membrane, and thus, came into direct contact with the luminal surface of the brain endothelium below (Supplementary Movie 2). These hypoxia-induced iPS-BMVECs also formed well developed tight junctions containing ZO-1 and Claudin-5, and expressed high levels of the cell–cell adhesion protein PECAM-1 along their lateral borders, as well as GLUT-1 and P-gp transporters on their

apical cell membrane (Fig. 1c). Electron microscopic analysis confirmed that the differentiated iPS-BMVEC monolayer also displayed well-developed tight junctions and adherens junctions with characteristic morphology along the membrane–membrane interfaces between adjacent endothelial cells (Fig. 1d). Under these flow conditions, the endothelial monolayer within the human BBB Chip maintained its tight junctional integrity and continued to express high levels of ZO-1 as well as the glucose transporter GLUT-1—the major nonneuronal glucose transporter in brain—on their apical surface for at least 1 week in culture, as detected by immunofluorescence microscopy (Fig. 1e).

Hypoxia-induced enhancement of brain-specific endothelial cell differentiation was also observed when gene and protein expression levels of representative BBB markers were compared with iPS-BMVECs cultured on-chip differentiated under hypoxic versus normoxic conditions. Statistically significant increases in mRNA expression were observed for genes encoding GLUT-1, insulin receptor protein (INSR), and the BBB efflux transport proteins P-gp, BCRP (breast cancer resistant protein), and multidrug resistance proteins 1 and 4 (MRP1 and MRP4), as well as endothelial cell-specific VE-cadherin (Supplementary Figure 2a and Supplementary Table 3). Increases in P-gp and BCRP expression were also confirmed by western blot analysis (Supplementary Figure 2b and Supplementary Table 3). More-over, mass spectrometric (MS) analysis of iPS-BMVECs collected from BBB Chips on the 3rd day of seeding revealed higher protein levels of ATP-binding cassette (ABC) and solute carrier (SLC) transporter proteins in hypoxia-differentiated brain endothelium cultured in the BBB Chips compared with endothelium differentiated under normoxic conditions (Fig. 2a, Supplementary Figure 3 and Supplementary Table 4). These results are similar to those from previous proteomic studies that also demonstrated enrichment of BCRP1, MRP1, and MRP4 proteins in the BBB relative to other microvessels in animal studies[30]. In addition, MS analysis revealed that the hypoxia-induced iPS-BMVECs deposit their own basement membrane ECM containing higher levels of collagen IV, laminin, and perlecan (basement membrane-specific heparan sulfate proteoglycan), fibronectin, SPARC, and agrin compared with when differentiated under normoxic conditions (Supplementary Table 5); these molecules are also found within the basement membrane of the human BBB in vivo[31]. Further-more, to determine the contribution of astrocytes and pericytes to the function of iPS-derived human brain microvascular endothelium, we compared the mRNA profiles of iPS-BMVECs in mono-culture versus co-culture under continuous flow in the microfluidic BBB Chip. The mRNA levels of INSR, VE-cadherin, MRP1, and ZO-1 were all significantly higher in iPS-derived microvascular endothelium when co-cultured with brain astro-cytes and pericytes on-chip (Supplementary Figure 4).

**Reconstitution of in vivo levels of BBB function on-chip.** Importantly, the higher expression of BMVEC-specific surface markers we observed in the presence of pericytes and astrocytes (Fig. 1, Supplementary Figure 4) was associated with enhanced barrier function, as quantified by measuring TEER on-chip using BBB Chips that contained integrated electrodes[32]. This method generates results as impedance in Ohms ($\Omega$) rather than resis-tance ($\Omega \cdot cm^2$) because it is not normalized for surface area. These studies revealed that the impedance values of BBB Chips created with iPS-BMVECs differentiated under hypoxic conditions along with astrocytes and pericytes were ~25,000 $\Omega$ (Fig. 2b), which is 2 orders of magnitude higher than levels previously reported for BBB chips containing primary human BMVECs along with the brain perivascular cells (~400 $\Omega$) (Supplementary Figure 5). In contrast, while chips containing iPS-BMVECs established under normoxic conditions displayed a similar high maximum

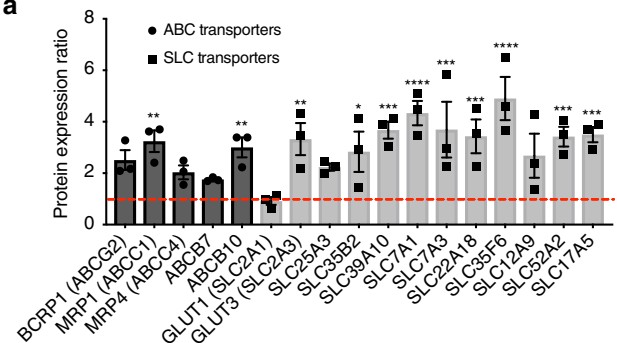

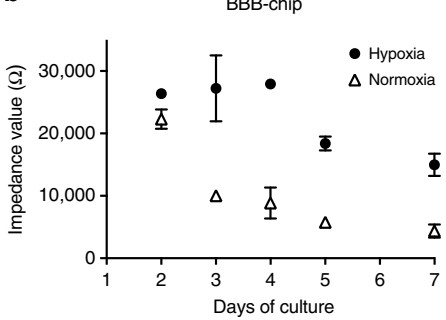

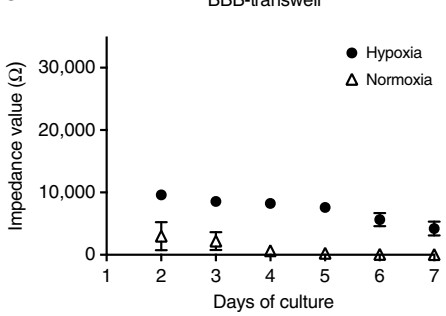

**Fig. 2** Physical barrier functions of the human BBB reconstituted on-chip. **a** Graph showing protein expression ratios of ABC and SLC transporter proteins in the iPS-hBMVECs differentiated under hypoxic conditions relative to normoxic conditions. **b** Impedance values for the human BBB measured in the TEER chip over the first 7 days of culture on-chip (recorded in the frequency range of 0.1 Hz–100 kHz) with iPS-BMVECs that were differentiated under hypoxia (closed circles) or normoxia (open triangles). **c** Impedance values for the BBB formed by the same human astrocytes, pericytes, and iPS-hBMVECs that were differentiated under hypoxia (closed circles) or normoxia (open triangles) but cultured in static Transwell inserts measured by TEER measurement machine (EVOM2). Data are represented as ± SE, $N = 3$. Statistical analysis is two-way ANOVA with Tukey's multiple comparisons test; *$P < 0.05$; **$P < 0.001$; ***$P < 0.0001$

impedance value of ~24,000 $\Omega$ at 2 days after plating, barrier levels dropped by more than 50% by day 3, and by >80% at 1 week (Fig. 2b). These latter results were nearly identical to those previously reported by others who cultured iPS-BMVECs estab-lished under normoxic conditions in either Transwell inserts or Organ Chip devices[13,33]. We independently confirmed these results using fluorescently labeled dextran tracers to measure barrier integrity. The apparent permeability ($P_{app}$) of the BBB Chip lined by hypoxia-induced iPS-BMVECs, astrocytes and pericytes was very low (~$10^{-8}$–$10^{-9}$ cm s$^{-1}$), and the $P_{app}$ value inversely correlated with the size of the tracer (average $P_{app} = $

~8.9, 1.1, and $0.24 \times 10^{-8}$ cm s$^{-1}$ for 3,10, and 70 kDa dextrans, respectively) (Supplementary Figure 6), as previously demonstrated in vitro and in vivo[34,35]. It is important to note that fluid flow also appeared to be required to sustain barrier function over time as we did not observe this level of barrier development when the same hypoxia-induced BMVECs were interfaced with astrocytes and pericytes cells in static Transwell cultures (Fig. 2c).

Furthermore, when we treated BBB Chips lined with iPS-BMVECs differentiated under normoxic conditions with the HIF1α inducer CoCl$_2$ (100 μM), we produced nearly identical enhancement of TEER and similar results were obtained with another chemical hypoxia mimetic, dimethyloxalylglycine (DMOG; 100 μM) (Supplementary Figure 7). The highly enhanced, in vivo-like level of barrier function ($> 20,000$ Ω) obtained with iPS-derived hBMVECs differentiated using CoCl$_2$ also could be prolonged for more than 2 weeks when cultured under continuous flow in the microfluidic BBB Chip (Supplementary Figure 8a). In contrast, this high level of barrier function again was not obtained when the same cells were interfaced with astrocytes and pericytes under static conditions in Transwell cultures (Supplementary Figure 8b). Confocal immunofluorescence microscopic analysis of the CoCl$_2$-activated human BBB Chips also confirmed that high levels of expression of ZO-1 and GLUT-1 were sustained on the surface of the iPS-BMVECs in these cultures (Supplementary Figure 9).

**An in vitro model to study molecular trafficking across the BBB.** Next, we tested if the enhanced human BBB Chip containing hypoxia-differentiated iPS-BMVECs, astrocytes and pericytes can recreate a functional metabolic barrier that regulates molecular traffic across the BBB, and if this occurs in a more physiologically relevant manner than in chips containing cells differentiated under normoxic conditions. The array of ABC transporters in the BBB, which includes P-gp, MRP1 & 4, and BCRP, provides an important defense mechanism that actively secretes molecules leaked out of blood back into the capillary lumen in order to protect the brain[36,37]. As we found that all of these ABC efflux pumps are expressed on the apical surface of the human iPS-BMVECs induced by hypoxia (Fig. 1c, Fig. 2a), we next analyzed their functionality and substrate selectivity. The hypoxia induction method was used for this and all subsequent studies because small molecule inducers may not mimic all of the effects of the hypoxic environment[38,39], and thus, their use would require further system characterization.

To assess P-gp activity in the BBB Chip, we pretreated the chips with the P-gp inhibitor, verapamil, and then the known P-gp substrates, rhodamine 123 and DiOC2[40] as well as the drug citalopram, were perfused through the endothelium-lined vascular channel of the BBB Chips. Transcytosis of the P-gp substrate molecules with and without P-gp inhibitor treatment was measured under the same continuous flow conditions (100 μL h$^{-1}$), and hence shear stress, in both channels; thus, there was no difference in the unstirred water layer between the chips. Transcytosis of the P-gp substrate molecules was also carried out under static conditions using the BBB Transwell model on the second day after the cell seeding when the barrier integrities were similar to the BBB Chip model (3 kDa Dextran $P_{app}$ values = ~$2 \times 10^{-7}$ cm s$^{-1}$). The $P_{app}$ of rhodamine 123 and DiOC2 were calculated by measuring changes in their fluorescent intensities between the two channels; MS analysis was used to measure the permeability of citalopram.

These studied revealed significant enhancement of the permeability of all three of these molecules through the BBB on-chip when the P-gp efflux pump was blocked by its inhibitor, verapamil, whereas a significant increase in the permeability of these molecules could not be detected when the same cells were

interfaced to form a BBB in static Transwell cultures (Fig. 3b). Remarkably, citalopram was also shown to increase its permeability when P-gp was inhibited in the microfluidic BBB Chip (Fig. 3a), which recapitulates similar in vivo findings shown in past animal studies[10,41]. This is an important result because all past in vitro BBB models failed to identify citalopram as a P-gp substrate[42,43].

To analyze the activities of BCRP and MRP1 transporter proteins, the BBB Chips were pretreated with inhibitors of various ABC transporter inhibitors, including verapamil MK571, and Ko143 that target P-gp, the MRPs and BCRP, respectively, and then uptake of fluorescent substrates, rhodamine 123 and DiOC2, was quantified under flow using fluorescent dextran (3 kDa) to monitor barrier integrity. When we measured the $P_{app}$ of these two ABC transporter substrates in BBB Chips containing iPS-BMVECs differentiated under hypoxic conditions along with astrocytes and pericytes, we found that inhibition of P-gp with verapamil resulted in significant influx of rhodamine 123 and DiOC2 into the CNS channel, and inhibition of BCRP with Ko143 had a small but significant effect on transport of DiOC2, but no effect on rhodamine 123, as previously described in other models[13,44] (Fig. 3c). In contrast, when similar studies were carried out with BBB Chips generated under normoxic conditions, we observed an increase of rhodamine 123 in the CNS channel when MRP1 was inhibited, but no change when P-gp was inhibited, and DiOC2 influx was not affected by any of the inhibitors (Fig. 3c). As P-gp is the main efflux pump for rhodamine 123[45,46] and DiOC2 influx was modulated in a manner similar to what is observed in vivo[45] in the hypoxic chips, these studies demonstrate that while the iPS-BMVECs differentiated under normoxic conditions may generate an effective permeability barrier[13,47], they do not fully recapitulate the human BBB's specialized molecular transport functions that are highly relevant to the drug development process.

To further investigate the potential of the hypoxia-induced human BBB Chip as an in vitro tool to study brain transporter-dependent drug efflux, we tested the cancer drug doxorubicin (Dox). While Dox is used to treat many types of cancer, its efficacy for brain tumors is limited due to poor penetration of the drug through the BBB because it is pumped back out of the endothelium and into the blood predominantly by P-gp transporters[48], and less by MRP1[49] or BCRP[50]. Consistent with these in vivo observations in humans, we found that inhibition of P-gp with Verapamil induced more than a 2.7-fold increase in Dox influx into the CNS channel (rise in $P_{app}$ from 1.4 to $3.7 \times 10^{-8}$ cm s$^{-1}$) when added under flow (100 μL h$^{-1}$) through the vascular channel of the BBB Chip (Fig. 3c), which is similar to the P-gp efflux ratio observed in vivo[37]. Again, as observed in vivo, inhibition of MRP1 or BCRP did not alter the permeation of Dox into the CNS channel (Fig. 3c). Together, these results suggest that differentiation of iPS-BMVECs in the presence of hypoxia, and positioning of these cells under physiological flow at a tissue-tissue interface with human brain pericytes and astrocytes, generates of an artificial human BBB that exhibits higher substrate specificity and functionality of the efflux barrier than previously reported in any in vitro BBB model.

**Reversible osmotic opening of the human BBB on-chip.** Hyperosmolar agents, such as high concentration mannitol solutions, are used clinically to overcome the BBB permeability limitation in order to increase delivery of drugs to the brain[51,52]. Intravenous infusion of a mannitol solution induces reversible opening of pores within the BBB with a radius of about 200 Å in animals[53] and humans[53,54], and this recovers due to bulk fluid flow from the blood to the brain[54]. We therefore explored whether we can use our hypoxia-enhanced BBB Chip to model

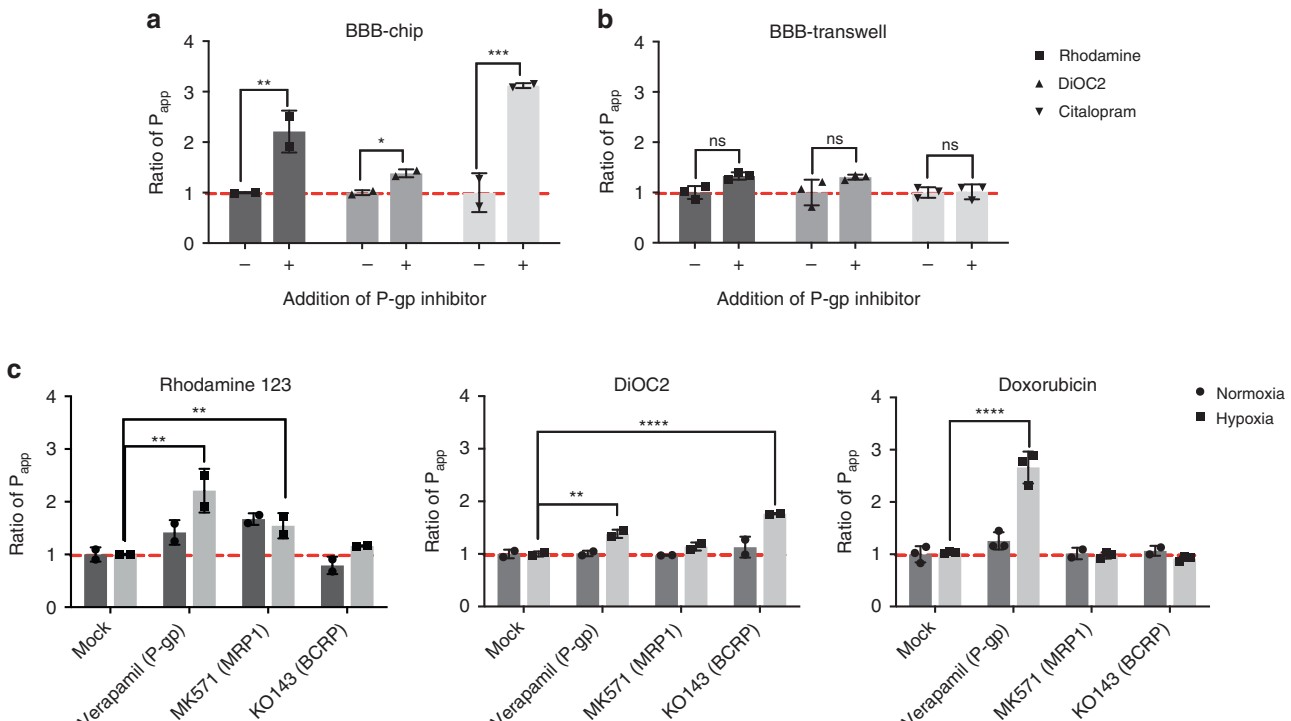

**Fig. 3** Metabolic barrier function of the human BBB Chip. P-gp activities measured in the BBB Chip (**a**) versus BBB Transwells (**b**) that were pretreated with verapamil to inhibit the P-gp activity in the presence or absence of the P-gp substrates, rhodamine 123, DiOC2, or citalopram. **c** BBB efflux pump substrate selectivity and functionality measured in the BBB Chips using rhodamine 123 as a substrate of P-gp and MRP1 (left); DiOC2 as a substrate of P-gp and BCRP (middle); and doxorubicin interacting with P-gp (right) with or without addition of efflux transporter inhibitors (verapamil for P-gp, MK 571 for MRPs, and Ko143 for BCRP). Ratio of $P_{app}$ indicates fold changes in the apparent permeability ($P_{app}$) of molecules resulting from chemical modulation of specific efflux pump activity. Data are presented as means ± SE, $N = 2$ for the chip experiments for rhodamine 123, DiOC2, and citalopram, as well as $N = 3$ for the BBB Transwell experiments and BBB chip experiments for Doxorubicin. The statistical analysis is two-way ANOVA with Tukey's multiple comparisons test; *$P < 0.05$; **$P < 0.001$; ***$P < 0.0001$.

delivery of large antibodies, such as those used as cancer therapeutics, by opening the barrier with hypertonic mannitol on-chip. Hypertonic (485 mOsmol L$^{-1}$) mannitol solution was flowed through the endothelium-lined vascular channel of the BBB Chip for 1 h to open the BBB, and then additional hypertonic medium containing 10 kDa dextran and the anti-cancer therapeutic antibody, cetuximab, was flowed for an additional hour through the same channel (Fig. 4a).

Constant exposure to the hypertonic medium resulted in a 50% drop in TEER within 2 h, which gradually recovered to normal levels within 4 h following infusion of isotonic medium into the vascular channel (Fig. 4b). Osmotic opening of the BBB in this manner also resulted in increased penetration of both 10 kDa dextran (Fig. 4c) and the cetuximab antibody (Fig. 4d) into the CNS channel, as measured by a rise in $P_{app}$ from 3 to 32 × 10$^{-8}$ cm s$^{-1}$ and from 2 to 5.1 × 10$^{-8}$ cm s$^{-1}$, respectively. The hypertonic medium was replaced by isotonic medium containing the same compounds after 1 h to allow the BBB Chip to recover barrier function, and after an additional 30 min, we found that the barrier integrity measured by TEER had not fully recovered (Fig. 4b), indicating that both the dextran and the antibody continued to penetrate through the BBB and into the CNS channel. However, the permeabilities of dextran (Fig. 4c) and the antibody (Fig. 4d) decreased to the normal range within 4 h, resulting in full recovery of barrier integrity (Fig. 4b). To our knowledge, this is the first demonstration of delivery of a clinically approved antibody drug by reversible osmotic opening of the human BBB in vitro, which mimics responses seen in human patients[52,54].

**Recapitulation of BBB-shuttling activities on-chip**. The BBB prevents the transport of nearly 100% of large drugs (> 500 Da) and antibodies (150 kDa) into the brain, and thus, many laboratories are seeking ways to discover new brain-targeting molecules that could enable delivery of drugs across the BBB and into the CNS. A few brain shuttling molecules have been identified, including IgG antibodies[55] and ligands for cell surface receptors on the brain microvascular endothelium that help drugs transit through the endothelium and into the CNS via receptor-mediated transcytosis[56]. Accurate in vitro assessment of CNS-targeted drug delivery has not been possible previously because it requires that human BBB culture models provide a physiological barrier that prevents paracellular leakage, as well as expression of brain endothelial cell-specific receptors and transporters. Given the high level of differentiation of iPS-BMVECs in our hypoxia-induced human BBB Chip, we explored whether this improved model can be used to directly assess the human BBB-penetrating capacity of peptides, nanoparticles, and antibodies in vitro.

Angiopep-2 is a small peptide ligand of LRP-1, expressed by iPS-BMVECs (Supplementary Figure 10), that has been shown to penetrate the BBB in vitro and in vivo[57]. The brain penetrating capacity of multiple drugs, including paclitaxel[58] and a cancer therapeutic HER-2 antibody[59], has been shown to be enhanced when modified with Angiopep-2. When we flowed Q-dot nanoparticles (20 nm) coated with Angiopep-2 through the endothelium-lined vascular channel of the human BBB Chip, these particles exhibited. Approximately 3.5-fold greater capacity to penetrate into the CNS channel compared with Q-dots

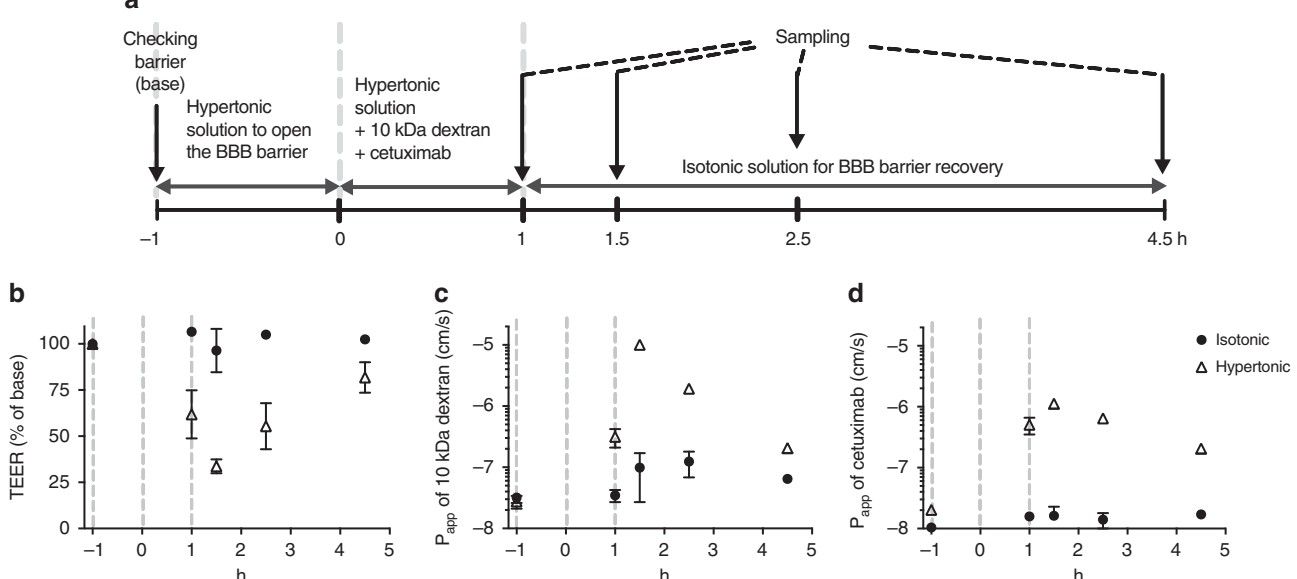

**Fig. 4** Penetration of a therapeutic antibody through the BBB Chip using osmotic solutions. **a** Experimental design for assessing osmotic opening of endothelial barrier in the BBB chip, including the timeline for medium (hypertonic and isotonic) changes, and drug and tracer dosing, as well as sampling times. Barrier integrity of the BBB Chips under isotonic and hypertonic conditions as monitored by measuring TEER (**b**) or $P_{app}$ of 10 kDa dextran-conjugated to tetramethylrhodamine (**c**) or of the therapeutic antibody cetuximab (**d**). Data are presented as means ± SE, $N = 2$ representing independent BBB Chips

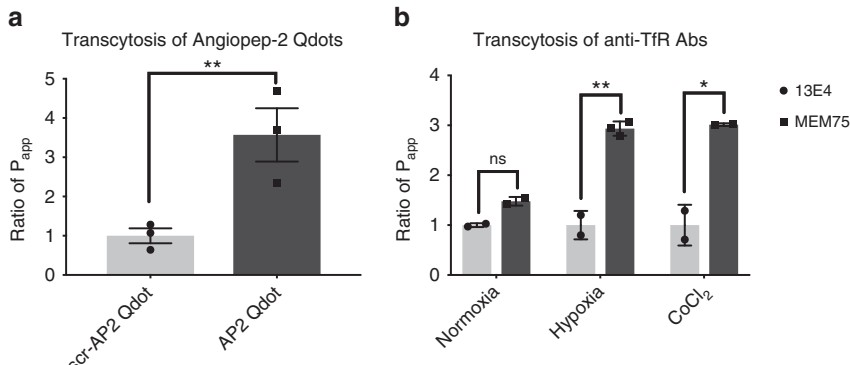

**Fig. 5** Recapitulation of Angiopep-2 peptide and anti-TfR antibody shuttling through the BBB on-chip. **a** Transcytosis of the known BBB shuttle molecule, Angiopep-2, measured by quantifying the Ratio of $P_{app}$ of Angiopep-2-conjugated quantum dot (Q-dot) nanoparticles versus Q-dots coated with a scrambled version of the Angiopep-2 peptide in chips lined by iPS-BMVECs differentiated under hypoxic conditions. $N = 3$, representing independent BBB Chips. **b** Ratio of $P_{app}$ of the known BBB-shuttling low-affinity anti-TfR antibody MEM75 (dark gray) relative to that displayed by the non-shuttling high-affinity anti-TfR 13E4 antibody (light gray) measured in human BBB Chips generated with iPS-BMVECs differentiated under normoxic or hypoxic conditions, or in the presence of $CoCl_2$. Data are presented as means ± SE, $N = 2$ representing independent BBB Chips. The statistical analysis is two-way ANOVA with Tukey's multiple comparisons test; *$P < 0.05$; **$P < 0.001$; ***$P < 0.0001$

coated with a scrambled peptide (Fig. 5a), even though there was no significant change in barrier integrity (Supplementary Figure 11a).

The transferrin receptor (TfR) on BMVECs (Supplementary Figure 10) also has been shown to function as a molecular shuttle[60] that can mediate differential penetration of anti-TfR antibodies through the BBB that differ in their binding affinity. More specifically, the anti-TfR antibody MEM75 has been shown to penetrate into the brain more efficiently than the 13E4 antibody in vitro[61]. This is because MEM75 has a reduced affinity for the receptor at pH 5.5, while the 13E4 anti-TfR antibody exhibits a pH-independent affinity that results in its entrapment within intracellular vesicles in brain endothelial cells. In past studies with in vitro human BBB models, including studies with

human iPS-BMVECs in Transwell cultures, it was not possible to replicate this highly clinically relevant form of shuttling across the BBB because of the poor barrier function and high levels of paracellular influx[61,62], and we obtained similar results when MEM75 or 13E4 antibodies were flowed through the vascular channel with the BBB Chip containing endothelium differentiated under normoxic conditions (Fig. 5b). We also did not observe any significant difference in transcytosis ability between these two TfR antibodies when we flowed the antibodies through BBB Chips lined by primary (as opposed to iPSC-derived) human BMVECs (Supplementary Figure 12), which produced only a minimal (~1.4-fold) difference in transcytosis abilities for the two TfR antibodies much as previously observed in past microfluidic primary BBB models[63]. In contrast, when these anti-TfR

antibodies were flowed through the vascular channel of the hypoxia-enhanced BBB Chip, we were able to demonstrate threefold higher penetrance of MEM75 into the CNS channel relative to the 13E4 antibodies that was previously predicted in vitro[61] (Fig. 5b), and this was accomplished without compromising general barrier integrity (Supplementary Figure 11b). In addition, BBB Chips formed with iPS-BMVECs induced by exposure to CoCl$_2$, exhibited similar differential transcellular transport, again capturing the threefold difference between the two anti-TFR antibodies (Fig. 5b) without disrupting barrier integrity (Supplementary Figure 11c).

## Discussion

Taken together, these results demonstrate that mimicking the hypoxic microenvironment of the developing brain during differentiation of human iPSCs into BMVECs, by either lowering oxygen levels or adding HIF1α-inducing mimetics (e.g., CoCl$_2$ or DMOG), enables differentiation of endothelial cells that recapitulate human-relevant physiological BBB properties when the cells are interfaced with human brain astrocytes and pericytes in a 2-channel microfluidic BBB Chip. This is the first human BBB Chip model that permits analysis of the BBB-penetrating activities of known BBB shuttle peptides, as well as analysis of the TfR-based antibody shuttling mechanism[61], under physiologically relevant flow conditions in vitro. In addition, this enhanced BBB Chip replicates the delivery of a clinically approved therapeutic antibody (cetuximab) by reversible osmotic opening of the human BBB in vitro, as is observed in human patients. Thus, while past work in this field described other human BBB models that contained similar cells maintained under different culture conditions (e.g., single channel microfluidic devices[64], 3D tube-like cultures[6,65]), none of them demonstrated these various clinically relevant functionalities (e.g., in vivo-like barrier function for 2 weeks compared with 2–3 days in past studies) and technical capabilities (e.g., real-time measurements of barrier function using integrated TEER electrodes). These functional properties are crucial for a preclinical drug development tool that can be used for discovery of new BBB shuttles and brain-targeted therapeutics, which was the specific focus of this study.

The human BBB Chip containing human iPS-BMVEC differentiated under developmentally inspired hypoxic conditions exhibited greatly increased and sustained barrier function than past in vitro BBB models with TEER levels similar to those estimated for the human BBB in vivo[66]. Moreover, this enhanced induction of brain-specific endothelial cell differentiation by hypoxia is mediated by HIF1α, and the hypoxia conditioning step can be circumvented by artificially increasing HIF1α levels using small molecules (CoCl$_2$ or DMOG) that have been previously shown to mimic the effects of hypoxia under normoxic conditions[39,67].

We and others have previously developed 3D human BBB Chip models using immortalized, primary, or iPS-derived BMVECs, but they either fail to generate in vivo-like barrier function or rapidly lose their phenotypic properties[6,13,68,69]. Another microfluidic BBB Chip was developed as a potential tool for drug testing that contains human iPS-BMVECs derived under normoxic conditions and cultured under continuous flow[35]; however, those cells were interfaced with rat (rather than human) astrocytes, there were no pericytes in that model, and active barrier function specifically mediated by active efflux pumps was not fully assessed. Another microfluidic BBB model was developed utilizing iPS-BMVECs that provides a tight barrier function for up to 6 days[64], but this model only contains a single microfluidic channel, which makes it difficult to carry out TEER measurements, explore the contributions of parenchymal cells, or

test for the transcytotic delivery of different drugs from capillary channels to brain side. A self-organized microvascular model of the BBB utilizing human iPS-BMVECs and human primary astrocytes and pericytes cultured within a fibrin gel has been developed[65]; however, utilization of fibrin gel makes it difficult to incorporate electrodes for TEER measurements to quantitatively and continuously assess barrier function with high sensitivity, as done in our 2-channel human BBB Chip. Diffusion of small molecules into the fibrin gel, and long residence time, also makes it difficult to accurately and quantitatively measure trans-BBB permeability values when using fluorescent tracers. The advantage of the hypoxia-enhanced BBB Chip in which the differentiated human brain microvascular endothelium also experiences dynamic fluid flow and shear stress that significantly influence BBB structure and function[70] is that it sustains in vivo-like barrier functions, as well as tightly regulated transcellular transport activities that are crucial for drug development studies, for much longer period of time (at least 2 weeks) compared with past human BBB models. The 2-channel design of the human BBB Chips also enables direct and independent access to both the parenchymal and vascular compartments, as well as real-time TEER measurements for quantitative assessment of barrier function, which greatly facilitates trans-BBB shuttling studies. Hence, the human BBB Chip should provide a more accurate model for assessing the ability of chemicals, CNS therapeutics, and drug targeting vehicles to pass through the human BBB and enter the brain.

The enhanced differentiation conditions we applied were inspired by the observation that the BBB forms in a hypoxic environment during embryological development[20,22,38], and this is accompanied by expression of Wnt7a/Wnt7b[25,26], as well as HIF1α[27,28]. Interestingly, when we differentiated the human iPSCs under similar hypoxic conditions, they expressed a similar developmental program including expression of Wnt7a and HIF1α, indicating that hypoxia triggers these changes in vitro as well. Importantly, this response was also accompanied by upregulation of various biomarkers associated with brain neovascularization including VEGF, GLUT-1, VE-cadherin, and PECAM during the differentiation of iPS-BMVECs, as well as improved formation of cell–cell junctions and membrane transporters that resulted in significantly improved BBB function that was stable over longer times than observed in past models. But induction of HIF1α appears to be the essential trigger for the generation of an enhanced in vivo-like BBB Chip as similar induction of iPS-BMVEC differentiation could be obtained by addition of small molecule inducers of HIF1α (CoCl$_2$ and DMOG). In fact, treatment with these hypoxia mimetics may prove to be a useful method to generate improved iPSC-derived BBB Chips without requiring a specialized hypoxic chamber. It is also interesting to note that this hypoxia-induced differentiation response appears to be specific for the human iPSCs as exposure to human primary adult BMVECs to hypoxia results in the opposite response (e.g., increased levels of HIF1α induce expression of VEGF, which leads to higher BBB permeability and edema)[71,72]. In contrast, in the hypoxia-induced iPS-BMVECs, this response was not observed likely because HIF1α levels decreased drastically when the differentiated hBMVECs were seeded in the chip.

Using the hypoxia-enhanced human BBB Chip, we were able to demonstrate that many clinically relevant responses of the BBB can be recapitulated in vitro. The first and most important is maintenance of sustained levels of low barrier permeability similar to those observed in human brain for more than 2 weeks in vitro. We also showed that the BBB Chip can be used to study reversible opening of the BBB using hypertonic solutions in vitro, as is done in vivo; this technique may be used to develop improved therapeutic delivery strategies in the future, as

demonstrated here by delivery of an FDA approved therapeutic antibody, mimicking a clinically relevant drug delivery strategy[52]. Even more importantly, the enhanced BBB chip was able to mimic transporter-mediated drug efflux including appropriate substrate specificity. Using our enhanced BBB Chip, we first recapitulated in vivo interaction of P-gp and citalopram[10,41] under physiological flow, which could not be reproduced under static conditions in past 3D models. Therefore, our BBB Chip model demonstrated the advantage of using microfluidic system to mimic in vivo cellular transport of molecules. Furthermore, in vivo shuttling of CNS-targeting peptides, nanoparticles, and antibodies across the BBB were recapitulated using enhanced BBB Chip, enabling examination of CNS-targeting drug delivery systems as well.

In summary, the human BBB Chip created using iPS-BMVECs exposed to developmentally inspired hypoxic differentiation conditions and cultured under physiological flow when interfaced with human brain pericytes and astrocytes exhibits enhanced functionalities relative to past human BBB models. These functionalities provide long-term improvements resulting in formation of a stable BBB with high, in vivo-like permeability restriction that lasts up to 2 weeks; high levels of expression of tight junction, SLC and ABC proteins; and proper function of efflux proteins, as well as drug, peptide, nanoparticle, and antibody transcytosis capabilities that are dependent on TfR and LRP1 surface proteins. This enhanced human BBB Chip provides a significant advance as it allows for improved drug screening by better recapitulating the in vivo environment of the BBB. It may therefore prove useful for development of drugs or delivery vehicles that selectively cross the BBB and target the CNS, as well as for modeling CNS diseases in vitro using patient-derived iPSCs to investigate improved brain therapies and advance personalized medicine.

## Methods

**Cell culture**. The human iPS cell line IMR90-4 (WiCell Research Institute) was propagated on tissue-culture plate that were coated with Matrigel (BD Biosciences) by using mTeSR1 medium (Stem Cell Technology) and maintained according to WiCell Feeder Independent Pluripotent Stem Cell Protocols provided by the WiCell Research Institute (http://www.wicell.org). Primary human astrocytes isolated from cerebral cortex were obtained from ScienCell and maintained in Astrocyte medium (ScienCell). Primary human brain pericytes were also obtained from ScienCell and maintained in Pericyte medium (ScienCell). The primary cells were used at passage 3–6.

**BMVEC differentiation**. Human iPS cells were differentiated to hBMVEC as previously described[33] with modification of $O_2$ conditions. Briefly, IMR90-4 iPSCs (WiCell Research Institute, Inc.) were dissociated using Accutase and seeded on 6-well plate coated with Matrigel at a concentration of $1.8 \times 10^5$ cells per well. Cells were cultured with mTeSR1 media for three days until the concentration reaches to $2.5 \times 10^5$ cells per well. The human iPSCs were differentiated to a mixed endothelial cells and neural progenitor cell culture by switching cells to unconditioned medium (UM) for 6 days (D0-D6). UM includes 392.5 mL DMEM/F12 (Invitrogen), 100 mL of Knockout Serum Replacement (KOSR) (Thermo Fisher Scientific), 5 mL of non-essential amino acids (Invitrogen), 2.5 mL of Glutamax (Invitrogen), and 3.5 μL of β-mercaptoethanol (Sigma). The endothelial cells were selectively expanded by switching to endothelial cell (EC) media supplemented with retinoic acid (RA) (D7-D8). For steady 5% $O_2$ conditions (from D0 to D9), cells were cultured in the incubator (Heracell™ 150i Tri-Gas Incubator, Thermo Scientific) flushing with a 5% $O_2$–5% $CO_2$–$N_2$ balance continuously. To analyze the mRNA expressions during the BMVEC differentiation, total RNA was extracted from cell culture using RNeasy Mini Kit (Promega) and cDNA was synthesized using Maxima First Strand cDNA synthesis Kit according to the instruction's protocol. The multiplex qPCR was performed by using TaqMan probes (Thermo Fisher Scientific) in a QuantStudio 7 Flex (Thermo Fisher Scientific). Label-free proteomics were carried as described previously using mass spectrometry analysis[73].

**Device fabrication**. The design of the human BBB-on-a-chip was modified from the previously reported human Small Airway Chip[74]. Parts for apical and basal channels were cast in polydimethylsiloxane (PDMS) (Sylgard 184, Ellsworth Adhesives) at a 10:1 ratio of base to curing agent in custom 3D printed Prototherm

molds (Proto Labs). The hollow microchannels were 2-cm long and 1-mm wide; top and bottom channels were 1 and 0.2 mm high, respectively. PDMS was degassed at −80 kPa until all bubbles were removed, and cured at 60 °C for at least 4 h. Twenty-micrometer thick, transparent, track etched, polyethylene terephthalate (PET) membranes with 0.4-μm perpendicular pores at a pore density of $4 \times 10^6$ pores cm$^{-2}$, were purchased from A.R. Brown and laser cut to size and to add ports. Membranes were bonded to PDMS using an epoxy silane method modified from Tang and Lee[75]. Briefly, PDMS parts and membranes were treated with $O_2$ plasma at 20 W for 45 s, $O_2$ gas flow 50 sccm to 0.80 mbar. PDMS parts were submerged in a 1% (3-Glycidyloxypropyl)trimethoxysilane solution (Sigma) and membranes were submerged in a 5% (3-Aminopropyl)triethoxysilane solution (Sigma) for 20 min at room temperature to bond silanes to the surface of the parts. Parts were rinsed in MilliQ filtered water and dried with compressed air. Parts were aligned and bonded by hand and placed in a 60 °C oven under 500 g weights for at least 12 h to anneal the bond. Chips were inspected for delamination, debris, or other defects before use.

**BBB reconstitution on a chip**. PDMS surface of the chips were activated with oxygen plasma treatment and the channels were coated with collagen IV (400 μg ml$^{-1}$) and fibronectin (100 μg ml$^{-1}$) overnight. Both channels of the chip were rinsed with PBS and then with astrocyte media before seeding cells. For coculturing astrocytes and pericytes in the brain channel of the BBB Chip, a density of $0.7 \times 10^6$ cells ml$^{-1}$ of human astrocyte and $0.3 \times 10^6$ cells ml$^{-1}$ of pericyte were mixed together in the astrocyte media and seeded on the apical channel of the chip, then incubated in the incubator for 1 h. To remove the access of the astrocyte and pericytes, channels of the chip were washed with EC + RA medium (endothelial medium with addition of fibroblast growth factor (Fgf) and retinoic acid (RA)[33]) and then $2.3 \times 10^7$ cells ml$^{-1}$ of iPS-BMVEC (D8) was seeded in the basal channel, and the device was flipped immediately to allow the BMVECs to adhere to the ECM-coated PET membrane. After 5 h incubation in the hypoxic incubator, the device was flipped back to let the rest of BMVECs sit on the bottom and sides of the channel to form a capillary lumen. Chips seeded with cells were maintained in the hypoxic incubator for 24 h, and then the BBB Chip was fed with EC medium deprived of Fgf and RA and transferred to the regular incubator (5% $O_2$–5% $CO_2$). On the second day of cell seeding on the microfluidic device, BBB Chips were attached to peristaltic pumps (microprocessor controlled dispensing pump from Ismatec), and EC medium was flowed through the channels (60–100 μL h$^{-1}$) to allow the BBB Chips adjust to flow conditions; then on the third day of seeding dosing studies were performed. For TEER measurements, we seeded astrocyte, pericyte, and iPS-BMVEC on TEER chips[32], and measured the four-point impedance using a PGstat128N from Metrohm Autolab BV as previously reported[32]. Apparent permeability ($P_{app}$) of the barrier was calculated by following a previously described method[73]. Briefly, 100 μg mL$^{-1}$ of dextran tracers were dosed through the vascular channels for a known period of time, and concentration of the dextran tracers in the outlet samples from both vascular and brain channels was determined by using BioTek (BioTek Instruments, Inc., Winooski, VT, USA). Then, following equation was used to calculate $P_{app}$:

$$P_{app} = \frac{V_r \times C_r}{A \times t \times \frac{(C_{d-out} \times V_d + C_r \times V_r)}{(V_d + V_r)}} \quad (1)$$

Here, $V_r$ is volume of receiving channel at time $t$, $V_d$ is volume of dosing channel at time $t$, $A$ is area of membrane which is 0.167 cm$^2$ in our chip model, $C_r$ is measured concentration of tracer in the receiving channel, and $C_{d-out}$ is measured concentration of tracer in the dosing channel effluent. Furthermore, we quantified the percent loss of the molecules into the chip materials with and without cells being present. We tracked the inlet dosing concentrations for both vascular and brain channels as well as outlet concentrations of both channels. Tracer loss (%) was monitored and accounted for during the experiment using the following formula:

$$\text{Tracer loss (\%)} = 100 \times \left(1 - \frac{(C_{d-out} \times V_d) + (C_r \times V_r)}{C_{d-in} \times V_d}\right) \quad (2)$$

Here, $C_{d-out}$ is the measured concentration of tracer in dosing channel effluent, $C_{d-in}$ is the concentration of dosing medium in inlet, $C_r$ is the measured concentration of tracer in receiving channel effluent. $V_d$ is the volume of dosing channel effluent at time $t$, and volume of receiving channel effluent at time $t$.

**Immunofluorescence microscopy**. BBB Chips were fixed with 4% paraformaldehyde in PBS for 10 min and then washed with PBS. Immunostaining was performed after permeabilization in PBS with 0.1% Triton X-100 (Sigma) and blocking for 30 min in 10% goat serum in PBS with 0.1% Triton X-100. Antibodies used in the study were listed in the Online Reporting Summary, which were 1:100 dilutions in 10% goat serum and incubated overnight on the BBB Chip at 4 °C. Fluorescently conjugated secondary antibodies with Alexa Fluor-488, Alexa Fluor-555, or Alexa Fluor-647 were then used when the primary antibodies are not conjugated. Conventional confocal imaging was carried out with a 405-laser diode, an Argon laser and a tunable white laser using a Leica SP5 X MP Inverted Laser Scanning Confocal Microscope with an ~25 Å water immersion objective and a Zeiss Axio Observer microscope. For electron microscopy images, BBB chips samples were embedded in LR white resin by following the protocol used in Harvard Medical School Electron Microscopy Facility. Briefly, the BBB Chip is

washed in 0.1 M cacodylate buffer and post-fixed with 1% osmium tetroxide ($OsO_4$)/1.5% potassium ferrocyanide ($KFeCN_6$) for 1 h, washed in water for three times and incubated in 1% aqueous uranyl acetate for 1 h followed by two washes in water and subsequent dehydration in grades of alcohol (10 min each; 50%, 70%, 90%, 2 × 10 min 100%). The samples were incubated in a 1:1 mixture of 100% ethanol and LR White (EMS), followed by two changes of pure LR White for 1 h followed by overnight incubation in pure LR white. The following day, the samples were placed in a new LR White resin in gelatin capsules (EMS) and polymerized at 60 °C for 48 h. Ultra-thin sections (about 80 nm) were cut on a Reichert Ultracut-S microtome, picked up on copper grids, stained with lead citrate and examined in a JEOL 1200EX Transmission electron microscope. Images were recorded with an AMT 2k CCD camera and saved as TIFF files.

**BBB opening using hypertonic solution.** Hypertonic solution was generated by mixing 20% Mannitol Injection USP (1100 mOsmol $L^{-1}$) and EC (280 mOsmol $L^{-1}$) media in 1:3 ratio considering the dilution after intravenous or intra-arterial injection in the body. The isotonic solution was generated by mixing PBS and EC media in 1:3 ratio. The apparent permeability ($P_{app}$) of fluorescently labeled 10 kDa dextran tracer (Thermo Fisher) and cetuximab (Selleckchem.com, USA) on BBB Chip was measured before treating hypertonic solution and rinsed with regular EC media. To open the barrier, the endothelial channel in the BBB Chip was fed with hypertonic solution at 100 µl $h^{-1}$ flow rate for 1 h (−1 to 0 h), and the media was switched to hypertonic solution including 100 µg $ml^{-1}$ of 10 kDa dextran and 10 µg $ml^{-1}$ of cetuximab (0–1 h). After collecting the effluents at 1 h, the media was switched to isotonic media with same dose of 10 kDa dextran and cetuximab to let the barrier recover. The effluent was collected at 30 min, 1.5 h, and 3.5 h after switching to isotonic media, and the $P_{app}$ of the 10 kDa dextran and cetuximab were analyzed by microplate reader and human IgG1 ELISA kit (Sigma).

**Metabolic barrier function.** P-gp, MRP, and BCRP functionality were assessed using rhodamine 123 (Sigma), DiOC2 (3,3′-diethyloxacarbocyanine Iodide) (Sigma), citalopram (Sigma), and doxorubicin (Sigma). Both channels were pre-treated with 50 µM verapamil (Sigma), 10 µM MK571 (Sigma), or 1 µM Ko143, which are inhibitors of P-gp, MRPs, or BCRP, respectively. At 30 min after pre-treatment with inhibitors, rhodamine 123 (2 µM), DiOC2 (2 µM), citalopram or doxorubicin (5 µM) in the presence or absence of inhibitor was dosed to brain endothelial channel with flow rate at 100 µl $h^{-1}$. To monitor the barrier integrity, 100 µg $ml^{-1}$ of 3 kDa dextran-cascade blue (Thermo Fisher) was dosed simultaneously. We collected apical and basal effluent for 6 h in the effluents were quantified by measuring fluorescent intensity. The fluorescence was measured at 485/530 nm, 482/497 nm, or 470/585 nm to quantify rhodamine 123, DiOC2, or doxorubicin, respectively, using a Synergy H1 microplate reader (BioTek, USA). Amount of citalopram in the apical and basal media was quantified using mass spectroscopy. The increase of BBB permeability of the drugs in the presence of inhibitor was presented as 'ratio of $P_{app}$'.

**BBB transcytosis of antibodies and peptide-modified Q-dots.** For the transcytosis assay, 10 µg $ml^{-1}$ of MEM-75 or 13E4 from Abcam (Cambridge, UK) was flowed to the brain endothelial channel at 60 µl $h^{-1}$ for 3 h, and the effluents were collected for the analysis. The detection and quantification of antibodies were performed by ELISA as described in previous report[61]. Briefly, 5 µg $mL^{-1}$ of goat-anti-mouse antibody (Jackson ImmunoResearch) was immobilized on a high binding 96-well plate in carbonate buffer pH 9.5 for overnight at 4 °C. Plate was washed with PBS with 0.1% Tween 20 for three times. 1% BSA in PBS was used to block the plate for 1 h at RT. Antibody samples collected from BBB Chips were incubated on the plate in the EC media for 2 h at RT. Then, an HRP-conjugated anti-mouse secondary antibody (Jackson ImmunoResearch) was used to detect the MEM75 and 13E4 antibodies on the plate. HRP enzyme on the secondary antibody was quantified using a luminescent substrate (Sigma). Standard curves for MEM75 and 13E4 antibodies was generated on the same plate to calculate the concentrations of the antibodies in the samples (Supplementary Figure 13). The luminescent intensity of the plates was measured using a Synergy H1 microplate reader (BioTek, USA), and the antibody concentrations in the samples from the brain and vascular channels of the BBB Chip was quantified to calculate the $P_{app}$ (cm $s^{-1}$).

To generate the Angiopep-2 modified Q-dot, C-terminally biotinylated Angiopep-2 (TFFYGGSRGKRNNFKTEEY) and scrambled Angiopep-2 (TGFKYRFGSKEGRNNYTEF) were purchased through the custom peptide synthesis service of Peptide 2.0 (USA), and Qdot 655 ITK Streptavidin Conjugate Kit was purchased from Thermo Fisher Scientific. Four micrometers of C-terminally biotinylated synthetic Angiopep-2 or scrambled Angiopep-2 was incubated with 0.4 µM of Q-dots for 30 min at 4 °C with agitation. The unbound peptides were cleared by using protein desalting column, 7K MWCO (Pierce) following instruction. Four micrometers of Angiopep-2 modified or scrambled Angiopep-2 modified Q-dot was dosed to brain endothelial channel at 100 µl $h^{-1}$ for 6 h, and the effluents were collected for analysis. The fluorescent intensity of effluent was measured using a Synergy H1 microplate reader (BioTek, USA) at 488/655 nm, and the Q-dot was quantified to calculate the $P_{app}$ (cm $s^{-1}$).

**Statistical analyses.** All data represent means (± SE) of distinct sample measurements ($N > 2$). The statistical analysis is two-way ANOVA with Tukey's multiple comparisons test; *$P < 0.05$; **$P < 0.001$; ***$P < 0.0001$. Prism 7 (GraphPad Software) was used for statistical analysis.

**Reporting summary.** Further information on research design is available in the Nature Research Reporting Summary linked to this article.

## Data availability
All data generated or analyzed during this study are included in this published article (and its supplementary information files), or are available from the corresponding author upon request.

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

## Acknowledgements

This research was supported by funding from the Wyss Institute for Biologically Inspired Engineering (to D.E.I.), Defense Advanced Research Projects Agency under Cooperative Agreement Number W911NF-12-2-0036 (to D.E.I), and National Research Foundation of Korea (NRF) grant (Korea, government, Ministry of Science and ICT; NRF-2018R1A5A1024340 and 2018K1A4A3A01063890 to T-E.P.), and Knut and Alice Wallenberg Foundation (WAF 2015-0178) (to A.H.). We also thank T. Ferrante for technical assistance, and L. Jin for artwork and technical illustration.

## Author contributions

T-E.P. and N.M. participated in the design and performance of all experiments and analyzed the data, working with D.E.I., who also supervised all work. R.H., R.M., E.H.F., and H.S. helped perform experiments. A.W. and R.P-B. helped design experiments, interpret data, and supervised all work. A.H. helped design experiments, performed imaging and proteomics experiments, and interpret data. O.H. and M.B. prepared microfluidic chips and helped measure TEER and interpret data. H.J.M. and L.C.G. provided scientific vision on the experiments related to reversible osmotic opening of the human BBB. H.W.S. and S.P.P. help develop protocols for the differentiation of stem cells. E.S. provided scientific supervision on the BBB model as well as access to training on the differentiation of stem cells to the brain endothelial cells. N.M., T-E.P., and D.E.I. prepared the paper with input from all others.

## Additional information

**Competing interests:** D.E.I. is a founder and holds equity in Emulate, Inc., and chairs its scientific advisory board. All other authors declare no competing interests.

