## [Peer Review File · Nature Communications]

Reviewers' comments:

Reviewer #1 (Remarks to the Author):

The manuscript by Park et al describes an in vitro microfluidic blood-brain barrier (BBB) model. The model is created using human iPS cell-derived brain vascular endothelial cells (hiPS-BMVECs) co-cultured with astrocytes and pericytes under hypoxic condition. The present model offers many advantages to recapitulate in vivo BBB characteristics. However, the reviewer has some comments and questions.

1) Regarding Figure 2a, the reviewer cannot find P-gp (ABCB1) in this figure. Why is this ABC transporter removed in Figure 2a?

2) Some results on transporter mRNA and protein expressions (Figure2a, Figure 3, Figs S1b, S2a, S3, S4) are represented as "ratio". However, the value may be dependent on the quantitative measures of mRNA and protein expression levels. Therefore, the quantitative absolute values of mRNA and protein on transporter and receptor expressions SHOULD be disclosed in the "Supplementary Information".

3) Please discuss on the effect of unstirred water layer on the drug permeability in the microfluidic system, because the discrepancy between Figure 3a and Figure 3b is not necessary dependent on the P-gp expression.

Reviewer #2 (Remarks to the Author):

The authors present an in vitro model of the blood-brain barrier (BBB) fabricated using iPS-derived human brain endothelial cells, primary human pericytes and astrocytes. Culture under hypoxia conditions is shown to produce short-term changes in protein expression and barrier properties, recapitulating certain aspects of the BBB. While the study reports interesting data regarding transport of several molecules through the platform presented, especially active transport by a variety of transporter proteins, this reviewer cannot recommend its publication in Nature Communications due to lack of sufficient novelty and methodological rigor. In particular:

1. Recapitulation of physiological BBB barrier properties has been previously achieved by a number of groups. The authors cite the work from Shuler's group (reference 41), which also presents a microfluidic BBB monolayer model cultured under flow with physiological permeability values, and make the point that one of the cells types used in that case was not human, but fail to cite the work by Kamm's group on BBB microvasculature on chip (Campisi et al., Biomaterials 2018; 180, 117-129), which uses human cells only (iPSC-derived endothelial cells, and primary pericytes and astrocytes, as in the present model), and arranged in an arguably more physiological 3D structure. The work by the Searson group, also a 3D model of the human BBB with physiological barrier properties, is cited (reference 77), but the authors fail to make a compelling argument as to why their model is more relevant and why a single microfluidic channel makes it difficult to test molecular transport, as stated in the text. The fact that hypoxia aids the development of BBB-like properties, while interesting, is also not novel, as the authors themselves state, and its short-term effect does not represent a significant improvement over existing platforms. From a fabrication perspective, there is also no particular novelty in the device used.

2. Comparison to BBB barrier function is done for the most part through assessment of permeability, yet the method and equation used to measure it is never reported, a critical issue when the order of magnitude calculated can vary widely depending on the numerical model used. Absolute values for a number of molecules are also never reported (e.g. multiple molecules from Figure 3), and the order of magnitude for a single molecule can change broadly (e.g. 3 kDa dextran is initially reported to have a permeability of 10^{-8} to then increase more than 3-fold to 10^{-7} in Figure S10). Figure S10a leads this reviewer to suspect that three repeats are not sufficient to capture relevant changes in permeability between conditions.

3. In the introduction, the authors mention ECM as one of the crucial factors in the BBB microenvironment that determine BBB function. Moreover, authors coated the PET membrane that

separate the 2 channels with collagen type IV and fibronectin, generating an ECM-coated porous membrane. However, the coating alone, mainly used for cell adhesion, and the collagen gel do not fully represent a physiologically-relevant environment. Authors are encouraged to demonstrate that cells in the system are producing ECM such as laminin and collagen, commonly deposited by endothelial cells, pericytes and astrocytes in a healthy state. Indeed, authors are encouraged to show at least immunofluorescence, Western Blot or PCR of these proteins, improve the physiological relevance of the model and compare conditions with/without pericytes and astrocytes.

4. Another important aspect of the physiologic microenvironment is the stiffness of the surrounding matrix, which is not captured in the present model and not discussed as an important factor.

5. The authors make statements about the ability of the platform to shuttle large molecules through transcytosis, but the mode of transport of any of the molecules tested is never determined e.g. by using specific inhibitors of transcytosis or paracellular transport. These studies would be necessary in order to validate the authors' claims

Other comments and questions:

1. The experimental details for the electron microscopy of the endothelial junctions seem to be missing. How are tight junctions distinguished from adherens junctions? The images seem somewhat arbitrary and not particularly illuminating.

2. Accurate details about the size of the vascular channel are missing or not well reported (e.g. Figure 1a).

3. Why is F-actin chosen to identify pericytes? F-actin is not a commonly used pericyte marker. Besides, the iPSC-derived brain endothelial cells and astrocytes also express F-actin. However, in Figure 1a, endothelial cells were identified only by nuclei and do not seem to express F-actin. Any reason for that? Please specify.

4. Although supplementary videos are referred in the manuscript, their interpretation might be difficult without a clear explanation, such as marker used and colour code. Please address this point including proper captions in the supplementary information.

5. Why is dextran perfused together with other molecules? It seems from certain parts of the text that dextran is directly conjugated to the molecules, but the methodology is not clear.

6. The PDMS chip is likely to absorb a large fraction of small molecules over the several hours of the experiment. How is this decrease in concentration accounted for?

7. qPCR data for ZO1 is missing from Figure S1, despite its use in ratio form in Figure S4?

8. Even though it might be precluded by the fabrication steps of this BBB-chip, it would be useful and more physiologic to have pericytes and astrocytes surrounding the entire wall of the vascular channel, or at least the bottom surface, making a 3-channel device.

9. In some of the plots of Papp when not expressed as a ratio, the axis should be labelled as 10^{-7} rather than simply -7 (etc.).

10. The manuscript is generally well written, but a number of typos are present (e.g. line 7, "associated") and a thorough edit is required.

RESPONSE TO REVIEWERS

Reviewer #1:

1. Regarding Figure 2a, the reviewer cannot find Pgp (ABCB1) in this figure. Why is this ABC transporter removed in Figure 2a?

Figure 2a shows proteomics data comparing the relative expression of SLC and ABC proteins of iPS-BMVECs differentiated under hypoxia versus normoxia. Pgp could not be detected using the proteomic approach, and thus, we used three different independent methods (immunostaining, qRT-PCR, and Western blot analysis) to analyze its expression under these conditions; these data are shown in **Figures 1c, S2a, and S2b**, respectively. Most importantly, we also demonstrated Pgp functionality in the BBB Chip using known Pgp substrates and inhibitors, as shown in **Fig. 3a**.

2. Some results on transporter mRNA and protein expressions (Figure2a, Figure 3, Figs S1b, S2a, S3, S4) are represented as “ratio”. However, the value may be dependent on the quantitative measures of mRNA and protein expression levels. Therefore, the quantitative absolute values of mRNA and protein on transporter and receptor expressions SHOULD be disclosed in the “Supplementary Information”.

As requested, the quantitative absolute values of the mRNA levels of the proteins are summarized in the **Supplementary Table 1, 2, and 3**. The protein abundance values for iPS-BMVECs differentiated under hypoxia and normoxia are now summarized in the **Supplementary Table 4**.

3. Please discuss on the effect of unstirred water layer on the drug permeability in the microfluidic system, because the discrepancy between Figure 3a and Figure 3b is not necessary dependent on the Pgp expression.

In **Figure 3a,b**, we clearly show that the effects on drug permeability in the microfluidic BBB Chip do depend on Pgp expression and function as we only observe penetration of Pgp substrates when we add specific Pgp inhibitors to the system. As the same flow rate (and hence shear stress) was applied to the BBB Chips under all experimental conditions, there is no difference in the unstirred water layer between these conditions. The different Pgp functions we detected in the Transwell are also not due to changes in the unstirred water layer as the permeability of tracer dyes is indeed similar in the static Transwells and dynamic BBB Chips at the time point we carried out these studies. We now explain this in the revised Results.

Reviewer #2:

1. Recapitulation of physiological BBB barrier properties has been previously achieved by a number of groups. The authors cite the work from Shuler’s group (reference 41), which also presents a microfluidic BBB monolayer model cultured under flow with physiological permeability values, and make the point that one of the cells types used in that case was not human, but fail to cite the work by Kamm’s group on BBB microvasculature on chip (Campisi et al., Biomaterials 2018; 180, 117129), which uses human cells only (iPSCderived endothelial cells, and primary pericytes and astrocytes, as in the present model), and arranged in an arguably more physiological 3D structure.

The work by the Searson group, also a 3D model of the human BBB with physiological barrier properties, is cited (reference 77), but the authors fail to make a compelling argument as to why their model is more relevant and why a single microfluidic channel makes it difficult to test molecular transport, as stated in the text. The fact that hypoxia aids the development of BBB-like properties, while interesting, is also not novel, as the authors themselves state, and its short term effect does not represent a significant improvement over existing platform. From a fabrication perspective, there is also no particular novelty in the device used.

While past work in this field described other models of the human blood-brain barrier (BBB) that contained similar cells maintained under different culture conditions, none of them demonstrated the various clinically relevant functionalities (e.g., *in vivo*-like barrier function for 2 weeks compared to 2-3 days in past studies) and technical capabilities (e.g., *real-time* measurements of barrier function using integrated TEER electrodes) that are crucial for a preclinical tool that can be used for drug development as well as discovery of new BBB shuttles and brain-targeted therapeutics, which is the specific focus of our work on this human BBB Chip. More specifically, this is the first human BBB model that mimics the ability to enhance delivery of a clinically approved antibody drug by the reversible osmotic opening of the human BBB *in vitro*, as is observed in human patients. It is also the first to replicate the ability of low affinity anti-transferrin receptor antibodies to be preferentially transported across the BBB, again as has been previously demonstrated *in vivo*. Thus, our human BBB Chip is more suitable for investigation of drug transport and shuttling mechanisms than any previously published BBB model, and hence this should be of great interest to many scientists at pharmaceutical and biotechnology companies as well as clinicians and basic researchers.

As requested, we now cite the Campisi manuscript in the revised manuscript; however, we explain that utilization of a fibrin gel in that model makes it difficult to incorporate electrodes for TEER measurements to quantitatively and continuously assess barrier function with high sensitivity, as we do in our study. Moreover, we also note that the diffusion of small molecules into the gel, and long residence time there, makes it difficult to accurately and quantitatively measure trans-BBB permeability values even when using fluorescent tracers and imaging analysis. While the presence of a single channel as used in the Searson paper allows for quantification of molecular uptake by the cells, it is impossible to analyze or quantify transport or shuttling of molecules or delivery vehicles across the BBB (from the vascular channel to brain interstitial space), which is one of the most important and clinically relevant features of our model for drug development and discovery of new BBB shuttles. A simple example of the novelty of our model is our ability to replicate the differential trans-BBB shuttling function of anti-transferrin receptor antibodies that differ in their affinity for the receptor, and previously demonstrated by Genentech. These types of trans-BBB shuttle experiments are specifically enabled by the 2-channel design of our BBB Chip, which provides direct and independent access to both the vascular and brain channels; this is not possible with a single channel model.

While it is known that hypoxia influences brain microvessel development during embryological development, our finding that utilization of hypoxia conditions for a short period during differentiation of human iPS cells to the BMVECs provides long-term improvement in barrier formation (including enhanced expression of tight junction, transporter, and efflux proteins, as well as *in vivo*-like permeability restriction) that lasts up to 2 weeks (compared to 2-3 days in past studies) is entirely novel. Furthermore, our BBB Chip design enables TEER measurements while the human brain microvascular endothelium is experiencing dynamic fluid flow and shear stress that are known to significantly influence BBB structure and function. We now more clearly explain all of these points in the Results and Discussion.

2. Comparison to BBB barrier function is done for the most part through assessment of permeability, yet the method and equation used to measure it is never reported, a critical issue when the order of magnitude calculated can vary widely depending on the numerical model used. Absolute values for a number of molecules are also never reported (e.g. multiple molecules from Figure 3), and the order of magnitude for a single molecule can change broadly (e.g. 3 kDa dextran is initially reported to have a permeability of 10^8 to then increase more than 3-fold to 10^7 in Figure S10). Figure S10a leads this reviewer to suspect that three repeats are not sufficient to capture relevant changes in permeability between conditions.

We apologize for our oversight in terms of not including the formula we used for calculating apparent permeability (P_{app}); this formula and method are now explicitly described in the Methods. We also now include absolute P_{app} values of the **Figure 3** in **Supplementary Table 7**.

Supplementary Figure S11 (previously **Figure S10**) provides the barrier integrity data showing no significant barrier permeability changes between BBB chips dosed with different types of antibodies or Q-dots; indicating that penetration of antibodies or angiopep-2 labeled Q-dots are based on targeting the receptor and not due to the opening of the barrier (Ulbrich et al. 2009; Yu et al. 2014; Régina et al. 2008). We have observed variability of the barrier integrity between independent experiments since the barrier formation is depending on various factors including passage number and maintaining the iPS cell source, the coating of the chips with ECM, coculturing with other primary parenchymal cells, and keeping the flow conditions throughout the experiment. However, this variability was always within the 2-4 fold difference of the P_{app} values of the dextran tracers. It is important to note that we monitored the barrier integrity of the BBB Chips using fluorescent dextran tracers throughout the course of the transport experiments, and we compared these results within each experiment. In this manner, we were able to compare the transcytosis abilities of shuttle antibodies and peptide directly under conditions in which we confirmed that paracellular permeability remained low and hence, the barrier integrity of the chips remained intact.

3. In the introduction, the authors mention ECM as one of the crucial factors in the BBB microenvironment that determine BBB function. Moreover, authors coated the PET membrane that separate the 2 channels with collagen type IV and fibronectin, generating an ECM coated porous membrane. However, the coating alone, mainly used for cell adhesion, and the collagen gel do not fully represent a physiologically relevant environment. Authors are encouraged to demonstrate that cells in the system are producing ECM such as laminin and collagen, commonly deposited by endothelial cells, pericytes and astrocytes in a healthy state. Indeed, authors are encouraged to show at least immunofluorescence, Western Blot or PCR of these proteins, improve the physiological relevance of the model and compare conditions with/without pericytes and astrocytes.

The iPS-BMVECs do indeed accumulate their own basement membrane containing collagen IV, laminin, and perlecan (basement membrane-specific heparan sulfate proteoglycan), as well as fibronectin, SPARC, and agrin ((Baeten and Akassoglou 2011; Almutairi et al. 2016)) as demonstrated by proteomics analysis. We now include these data in **Supplementary Table S5**, and discuss this finding in the revised Results section.

4. Another important aspect of the physiologic microenvironment is the stiffness of the surrounding matrix, which is not captured in the present model and not discussed as an important factor.

As described in our response to point 3 above, the cells deposit their own basement membrane, which is likely more flexible than the ECM we originally immobilized on the substrate. But more importantly, it is known that fluidic shear plays an important role in driving BBB development, as described in Cucullo et al., BMC Neuroscience 2011, 12:40. We now explain these points in the Discussion and we added this citation.

5. The authors make statements about the ability of the platform to shuttle large molecules through transcytosis, but the mode of transport of any of the molecules tested is never determined e.g. by using specific inhibitors of transcytosis or paracellular transport. These studies would be necessary in order to validate the authors' claims.

We clearly demonstrated that anti-transferrin receptor (anti-TfR) antibodies, which are 'large molecules', as well as larger quantum dots (Q-dots) coated with peptides, are shuttled across the BBB under conditions where small molecular dextran tracers continued to be restricted passage, and hence, there is no rise in paracellular transport. Thus, we clearly demonstrate transcytosis of large molecules. Angiopep-2 also has been previously shown to target the LRP-1 protein that is expressed in the BBB Chips, which mediates its transcellular transport via transcytosis (Demeule et al. 2008)). Now we have included new Western Blot data confirming the expression of LRP-1 in the **Supplementary Figure S10**. In the Angiopep-2 transport experiments, we also used same size (20 nm) Q-dots without any peptide conjugation as a control, and the Angiopep-2 conjugated Q-dots could penetrate the BBB ~200 fold more efficiently than the non-targeted Q-dots. Similarly, anti-TfR antibodies are known to be targeting the TfR, which is expressed in the iPS-BMVECs as confirmed by Western Blot analysis (**Supplementary Figure S10**). For the antibody shuttling experiments, we selected two clones of anti-TfR antibodies (IgG) from the literature, which recapitulated the efficiency of the transport differences between those two antibodies that target the same receptor with different binding affinities depending on the pH (Sade et al. 2014). Consequently, we have clearly demonstrated that at least these two classes of trans-BBB shuttles that are known to be transported across the BBB via transcytosis are similarly transported in our model, while smaller molecular tracers that monitor changes in paracellular transport are not.

6. The experimental details for the electron microscopy of the endothelial junctions seem to be missing. How are tight junctions distinguished from adherens junctions? The images seem somewhat arbitrary and not particularly illuminating.

Adherens junctions (zonula adherens) and tight junctions (zonula occludens) were originally defined by the Nobel prize winner, George Palade, based entirely on their morphological appearance in electron microscopic images very similar to the one included here. Anyone skilled in the art can easily recognize them from the images we include in **Figure 1**. Most importantly, we confirm that these junctions are indeed present in the same locations in the endothelium by immunostaining for ZO-1, and that they are functional by demonstrating the extremely high and sustained, *in vivo*-like, barrier function of the BBB chip for weeks in culture. Although standard methods for transmission electron microscopy that are common in any lab that does this work were used in this study, we now include a more detailed description of these methods in the Methods section.

7. Accurate details about the size of the vascular channel are missing or not well reported (e.g. Figure 1a).

We apologize for this oversight; we now describe the size details of the microchannels in the Methods.

8. Why is F actin chosen to identify pericytes? F actin is not a commonly used pericyte marker. Besides, the iPSC-derived brain endothelial cells and astrocytes also express F actin. However, in Figure 1a, endothelial cells were identified only by nuclei and do not seem to express F actin. Any reason for that? Please specify.

F-actin is a common biomarker to identify pericytes (Campisi et al. 2018; Linville et al. 2018) and although it is present in all cell types, astrocytes and pericytes exhibit a different morphology and pattern of staining. We also used GFAP staining to discriminate astrocytes from pericytes, which enables us to identify pericytes in co-culture of the brain channel of the chip since pericytes are only positive for F-actin while being astrocytes positive for both F-actin and GFAP. Because we use a 2-channel chip, we have the advantage of being able to access and visualize cells within the vascular and brain channels of the chip individually. As there was only one cell type (endothelial cell) in the basal channel, we only stained them with ZO-1 to visualize tight junctions in **Figure 1a**; F-actin was not included to eliminate background.

9. Although supplementary videos are referred in the manuscript, their interpretation might be difficult without a clear explanation, such as marker used and colour code. Please address this point including proper captions in the supplementary information.

We now include descriptions of the color codes in the figure legends of the Supplementary videos in the Supplementary Information.

10. Why is dextran perfused together with other molecules? It seems from certain parts of the text that dextran is directly conjugated to the molecules, but the methodology is not clear.

Dextran was NOT conjugated to other molecules analyzed in this study. Instead, dextran molecules containing a fluorescent tag were used as tracers to monitor barrier integrity during the dosing period to ensure that there was no change in paracellular permeability or physical disruption on the barrier. This is an important control for studies designed to determine whether changes in trans-BBB transport are mediated by transcytosis, as described above. Protocol details of the permeability assay are described in the Methods.

11. The PDMS chip is likely to absorb a large fraction of small molecules over the several hours of the experiment. How is this decrease in concentration accounted for?

Another advantage of the 2-channel chip design is that we can quantify the percent loss of the molecules into the chip materials with or without cells being present. We track the inlet dosing concentrations for both vascular and brain channels as well as outlet concentrations of both channels. Tracer loss (%) is monitored and accounted for during the experiment using the following formula:

$$\text{Tracer loss (\%)} = 100 \times \left(1 - \frac{(C_{d-out} \times V_d) + (C_r \times V_r)}{C_{d-in} \times V_d} \right)$$

Here, C_{d-out} is the measured concentration of tracer in dosing channel effluent, C_{d-in} is the concentration of dosing medium in inlet, C_r is the measured concentration of tracer in receiving channel effluent. V_d is the volume of dosing channel effluent at time t , and volume of receiving channel effluent at time t . We now explain this in the Methods.

12. qPCR data for ZO1 is missing from Figure S1, despite its use in ratio form in Figure S4?

Figure S1 and **Figure S4** describe mRNA levels in cells at different stages of development, and they are shown for different purposes. **Figure S1** focuses on the time period of when iPS cells are induced to differentiate into BMVECs prior to initiation of chip studies, and we did not monitor mRNA levels for ZO1 during this time because it was not central to the question being asked. However, we do show these data in **Figure S4** because we are comparing the level of differentiation in iPS-BMVECs cultured on the microfluidic chips with and without addition of astrocytes and pericytes. We now note this in the Discussion.

13. Even though it might be precluded by the fabrication steps of this BBB chip, it would be useful and more physiologic to have pericytes and astrocytes surrounding the entire wall of the vascular channel, or at least the bottom surface, making a 3channel device.

As demonstrated in the **Supplementary Video 2**, the porous membrane of the chip design allows direct cell-cell contact formation between astrocytes, pericytes, and brain endothelial cells. Importantly, having only two channels (one vascular channel and one brain channel) greatly simplifies all calculations for molecule transport while still allowing screening for delivery vehicles that can cross the BBB and target the CNS, which is the major goal of this study. This would be very difficult using a 3-channel device. We now clarify these points in the Discussion.

14. In some of the plots of Papp when not expressed as a ratio, the axis should be labelled as 10^7 rather than simply 7 (etc.).

The axis labels in **Figure 4** were changed according to the Reviewers' suggestions:

15. The manuscript is generally well written, but a number of typos are present (e.g. line 7, "associated") and a thorough edit is required.

Thank you for this helpful input. We corrected all typos.

REVIEWERS' COMMENTS:

Reviewer #1 (Remarks to the Author):

The authors clearly respond to my questions and comments.

Reviewer #2 (Remarks to the Author):

I am satisfied with the way the authors answered my comments. I do advise the authors to edit the manuscript again for typos, as I could find many in the new sections added. The permeability formula given is also incorrect, as in the denominator the sign should be a minus and not a plus to define the concentration difference between the two channels.

RESPONSE TO REVIEWERS

Reviewer #1:

The authors clearly respond to my questions and comments.

We thank for the Reviewer's input which greatly improved the quality of our manuscript.

Reviewer #2:

I am satisfied with the way the authors answered my comments. I do advise the authors to edit the manuscript again for typos, as I could find many in the new sections added. The permeability formula given is also incorrect, as in the denominator the sign should be a minus and not a plus to define the concentration difference between the two channels.

We have corrected typographical errors throughout the manuscript. The permeability equation that we used is correct, as previously described (Maoz et al., *Nature Biotech.*, 2018). This is because the formula calculates the ratio of the amount of molecules in the receiving channel (calculated in the nominator) over the dosing amount of molecules which is the sum of both receiving and dosing channels' effluents while taking into account time and surface area (calculated in the denominator).